# Divide-and-Conquer Learning by Anchoring a Conical Hull

**Tianyi Zhou**[†]**, Jeff Bilmes**[‡]**, Carlos Guestrin**[†]
[†]Computer Science & Engineering, [‡]Electrical Engineering, University of Washington, Seattle
{tianyizh, bilmes, guestrin}@u.washington.edu

## Abstract

We reduce a broad class of fundamental machine learning problems, usually addressed by EM or sampling, to the problem of finding the $k$ extreme rays spanning the conical hull of a1 data point set. These $k$ "anchors" lead to a global solution and a more interpretable model that can even outperform EM and sampling on generalization error. To find the $k$ anchors, we propose a novel divide-and-conquer learning scheme "DCA" that distributes the problem to $\mathcal{O}(k \log k)$ same-type sub-problems on different low-D random hyperplanes, each can be solved independently by any existing solver. For the 2D sub-problem, we instead present a non-iterative solver that only needs to compute an array of cosine values and its max/min entries. DCA also provides a faster subroutine inside other algorithms to check whether a point is covered in a conical hull, and thus improves these algorithms by providing significant speedups. We apply our method to GMM, HMM, LDA, NMF and subspace clustering, then show its competitive performance and scalability over other methods on large datasets.

## 1 Introduction

Expectation-maximization (EM) [10], sampling methods [13], and matrix factorization [20, 25] are three algorithms commonly used to produce maximum likelihood (or maximum a posteriori (MAP)) estimates of models with latent variables/factors, and thus are used in a wide range of applications such as clustering, topic modeling, collaborative filtering, structured prediction, feature engineering, and time series analysis. However, their learning procedures rely on alternating optimization/updates between parameters and latent variables, a process that suffers from local optima. Hence, their quality greatly depends on initialization and on using a large number of iterations for proper convergence [24].

The method of moments [22, 6, 17], by contrast, solves $m$ equations by relating the first $m$ moments of observation $x \in \mathbb{R}^p$ to the $m$ model parameters, and thus yields a consistent estimator with a global solution. In practice, however, sample moments usually suffer from unbearably large variance, which easily leads to the failure of final estimation, especially when $m$ or $p$ is large. Although recent spectral methods [8, 18, 15, 1] reduces $m$ to 2 or 3 when estimating $\mathcal{O}(p) \gg m$ parameters [2] by relating the eigenspace of lower-order moments to parameters in a matrix form up to column scale, the variance of sample moments is still sensitive to large $p$ or data noise, which may result in poor estimation. Moreover, although spectral methods using SVDs or tensor decomposition evidently simplifies learning, the computation can still be expensive for big data. In addition, recovering a parameter matrix with uncertain column scale might not be feasible for some applications.

In this paper, we reduce the learning in a rich class of models (e.g., matrix factorization and latent variable model) to finding the extreme rays of a conical hull from a finite set of real data points. This is obtained by applying a general *separability assumption* to either the data matrix in matrix factorization or the $2^{nd}/3^{rd}$ order moments in latent variable models. Separability posits that a ground set of $n$ points, as rows of matrix $X$, can be represented by $X = FX_A$, where the rows (bases) in $X_A$ are a subset $A \subset V = [n]$ of rows in $X$, which are called "anchors" and are interesting to various

models when $|A| = k \ll n$. This property was introduced in [11] to establish the uniqueness of non-negative matrix factorization (NMF) under simplex constraints, and was later [19, 14] extended to non-negative constraints. We generalize it further to the model $X = FY_A$ for two (possibly distinct) finite sets of points $X$ and $Y$, and build a new theory for the identifiability of $A$. This generalization enables us to apply it to more general models (ref. Table 1) besides NMF. More interestingly, it leads to a learning method with much higher tolerance to the variance of sample moments or data noise, a unique global solution, and a more interpretable model.

Another primary contribution of this paper is a distributed learning scheme "divide-and-conquer anchoring" (DCA), for finding an anchor set $A$ such that $X = FY_A$ by solving same-type sub-problems on only $\mathcal{O}(k \log k)$ randomly drawn low-dimensional (low-D) hyperplanes. Each sub-problem is of the form of $(X\Phi) = F \cdot (Y\Phi)_A$ with random projection matrix $\Phi$, and can easily be handled by most solvers due to the low dimension. This is based on the observation that the geometry of the original conical hull is partially preserved after a random projection. We analyze the probability of success for each sub-problem to recover part of $A$, and then study the number of sub-problems

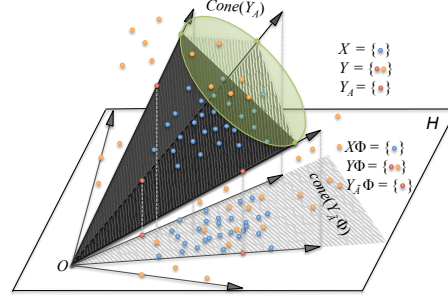

Figure 1: Geometry of general minimum conical hull problem and basic idea of divide-and-conquer anchoring (DCA).

for recovering the whole $A$ with high probability (*w.h.p.*). In particular, we propose an very fast non-iterative solver for sub-problems on the 2D plane, which requires computing an array of cosines and its max/min values, and thus results in learning algorithms with speedups of tens to hundreds of times. DCA improves multiple aspects of algorithm design since: 1) its idea of divide-and-conquer randomization gives rise to distributed learning that can reduce the original problem to multiple extremely low-D sub-problems that are much easier and faster to solve, and 2) it provides a fast subroutine, checking if a point is covered by a conical hull, which can be embedded into other solvers.

We apply both the conical hull anchoring model and DCA to five learning models: Gaussian mixture models (GMM) [27], hidden Markov models (HMM) [5], latent Dirichlet allocation (LDA) [7], NMF [20], and subspace clustering (SC) [12]. The resulting models and algorithms show significant improvement in efficiency. On generalization performance, they consistently outperform spectral methods and matrix factorization, and are comparable to or even better than EM and sampling.

In the following, we will first generalize the separability assumption and minimum conical hull problem risen from NMF in §2, and then show how to reduce more general learning models to a (general) minimum conical hull problem in §3. §4 presents a divide-and-conquer learning scheme that can quickly locate the anchors of the conical hull by solving the same problem in multiple extremely low-D spaces. Comprehensive experiments and comparison can be found in §5.

## 2  General Separability Assumption and Minimum Conical Hull Problem

The original separability property [11] is defined on the convex hull of a set of data points, namely that each point can be represented as a convex combination of certain subsets of vertices that define the convex hull. Later works on separable NMF [19, 14] extend it to the conical hull case, which replaced convex with conical combinations. Given the definition of (convex) cone and conical hull, the separability assumption can be defined both geometrically and algebraically.

**Definition 1** (**Cone & conical hull**). *A (convex) cone is a non-empty convex set that is closed with respect to conical combinations of its elements. In particular,* $\mathrm{cone}(R)$ *can be defined by its* $k$ *generators (or rays)* $R = \{r_i\}_{i=1}^k$ *such that*

$$\mathrm{cone}(R) = \left\{ \sum\nolimits_{i=1}^{k} \alpha_i r_i \mid r_i \in R, \alpha_i \in \mathbb{R}_+ \, \forall i \right\}. \tag{1}$$

See [29] for the original separability assumption, the equivalence between separable NMF and the minimum conical hull problem, which is defined as a submodular set cover problem.

### 2.1  General Separability Assumption and General Minimum Conical Hull Problem

By generalizing the separability assumption, we obtain a general minimum conical hull problem that can reduce more general learning models besides NMF, e.g., latent variable models and matrix factorization, to finding a set of "anchors" on the extreme rays of a conical hull.

**Definition 2** (**General separability assumption**). *All the $n$ data points(rows) in $X$ are covered in a finitely generated and pointed cone (i.e., if $x \in \text{cone}(Y_A)$ then $-x \notin \text{cone}(Y_A)$) whose generators form a subset $A \subseteq [m]$ of data points in $Y$ such that $\nexists i \neq j, Y_{A_i} = a \cdot Y_{A_j}$. Geometrically, it says*

$$\forall i \in [n], X_i \in \text{cone}\,(Y_A)\,, Y_A = \{y_i\}_{i \in A}. \tag{2}$$

*An equivalent algebraic form is $X = FY_A$, where $|A| = k, F' \in S \subseteq \mathbb{R}_+^{(n-k) \times k}$.*

When $X = Y$ and $S = \mathbb{R}_+^{(n-k) \times k}$, it degenerates to the original separability assumption given in [29]. We generalize the minimum conical hull problem from [29]. Under the general separability assumption, it aims to find the anchor set $A$ from the points in $Y$ rather than $X$.

**Definition 3** (**General Minimum Conical Hull Problem**). *Given a finite set of points $X$ and a set $Y$ having an index set $V = [m]$ of its rows, the general minimum conical hull problem finds the subset of rows in $Y$ that define a super-cone for all the rows in $X$. That is, find $A \in 2^V$ that solves:*

$$\min_{A \subset V} |A|, \; s.t., \; \text{cone}(Y_A) \supseteq \text{cone}(X). \tag{3}$$

*where $\text{cone}(Y_A)$ is the cone induced by the rows $A$ of $Y$.*

When $X = Y$, this also degenerates to the original minimum conical hull problem defined in [29]. A critical question is whether/when the solution $A$ is unique. When $X = Y$ and $X = FX_A$, by following the analysis of the separability assumption in [29],we can prove that $A$ is unique and identifiable given $X$. However, when $X \neq Y$ and $X = FY_A$, it is clear that there could be multiple legal choices of $A$ (e.g., there could be multiple layers of conical hulls containing a conical hull covering all points in $X$). Fortunately, when the rows of $Y$ are rank-one matrices after vectorization (concatenating all columns to a long vector), which is the common case in most latent variable models in §3.2, $A$ can be uniquely determined if the number of rows in $X$ exceeds 2.

**Lemma 1** (**Identifiability**). *If $X = FY_A$ with the additional structure $Y_s = \text{vec}(O_i^s \otimes O_j^s)$ where $O_i$ is a $p_i \times k$ matrix and $O_i^s$ is its $s^{th}$ column, under the general separability assumption in Definition 2, two (non-identical) rows in $X$ are sufficient to exactly recover the unique $A$, $O_i$ and $O_j$.*

See [29] for proof and additional uniqueness conditions when applied to latent variable models.

## 3 Minimum Conical Hull Problem for General Learning Models

Table 1: Summary of reducing NMF, SC, GMM, HMM and LDA to a conical hull anchoring model $X = FY_A$ in §3, and their learning algorithms achieved by $A = \text{DCA}(X, Y, k, \mathbb{M})$ in Algorithm 1 . Minimal conical hull $\tilde{A} = \text{MCH}(X, Y)$ is defined in Definition 4. $\text{vec}(\cdot)$ denotes the vectorization of a matrix. For GMM and HMM, $X_i \in \mathbb{R}^{n \times p_i}$ is the data matrix for view $i$ (i.e., a subset of features) and the $i^{th}$ observation of all triples of sequential observations, respectively. $X_{t,i}$ is the $t^{th}$ row of $X_i$ and associates with point/triple $t$. $\eta_t$ is a vector uniformly drawn from the unit sphere. More details are given in [29].

| Model | $X$ in conical hull problem | $Y$ in conical hull problem | $k$ in conical hull problem |
|---|---|---|---|
| NMF | data matrix $X \in \mathbb{R}_+^{n \times p}$ | $Y := X$ | # of factors |
| SC | data matrix $X \in \mathbb{R}^{n \times p}$ | $Y := X$ | # of basis from all clusters |
| GMM | $[\text{vec}[X_1^T X_2]; \text{vec}[X_1^T \text{Diag}(X_3 \eta_t) X_2]_{t \in [q]}]/n$ | $[\text{vec}(X_{t,1} \otimes X_{t,2})]_{t \in [n]}$ | # of components/clusters |
| HMM | $[\text{vec}[X_2^T X_3]; \text{vec}[X_2^T \text{Diag}(X_1 \eta_t) X_3]_{t \in [q]}]/n$ | $[\text{vec}(X_{t,2} \otimes X_{t,3})]_{t \in [n]}$ | # of hidden states |
| LDA | word-word co-occurrence matrix $X \in \mathbb{R}_+^{p \times p}$ | $Y := X$ | # of topics |

| Algo | Each sub-problem in DCA | Post-processing after $A := \bigcup_i \tilde{A}^i$ | Interpretation of anchors indexed by $A$ |
|---|---|---|---|
| NMF | $\tilde{A} = \text{MCH}(X\Phi, X\Phi)$, can be solved by (10) | solving $F$ in $X = FX_A$ | basis $X_A$ are real data points |
| SC | $\tilde{A} =$ anchors of clusters achieved by *meanshift*($\widehat{(X\Phi)\varphi}$) | clustering anchors $X_A$ | cluster $i$ is a cone $cone(X_{A_i})$ |
| GMM | $\tilde{A} = \text{MCH}(X\Phi, Y\Phi)$, can be solved by (10) | N/A | centers $[X_{A,i}]_{i \in [3]}$ from real data |
| HMM | $\tilde{A} = \text{MCH}(X\Phi, Y\Phi)$, can be solved by (10) | solving $T$ in $OT = X_{A,3}$ | emission matrix $O = X_{A,2}$ |
| LDA | $\tilde{A} = \text{MCH}(X\Phi, X\Phi)$, can be solved by (10) | col-normalize $\{F : X = FX_A\}$ | anchor word for topic $i$ (topic prob. $F_i$) |

In this section, we discuss how to reduce the learning of general models such as matrix factorization and latent variable models to the (general) minimum conical hull problem. Five examples are given in Table 1 to show how this general technique can be applied to specific models.

### 3.1 Matrix Factorization

Besides NMF, we consider more general matrix factorization (MF) models that can operate on negative features and specify a complicated structure of $F$. The MF $X = FW$ is a deterministic latent variable model where $F$ and $W$ are deterministic latent factors. By assigning a likelihood $p(X_{i,j}|F_i, (W^T)_j)$ and priors $p(F)$ and $p(W)$, its optimization model can be derived from maximum

likelihood or MAP estimate. The resulting objective is usually a loss function $\ell(\cdot)$ of $X - FW$ plus regularization terms for $F$ and $W$, i.e., $\min \ell(X, FW) + R_F(F) + R_W(W)$.

Similar to separable NMF, minimizing the objective of general MF can be reduced to a minimum conical hull problem that selects the subset $A$ with $X = FX_A$. In this setting, $R_W(W) = \sum_{i=1}^{k} g(W_i)$ where $g(w) = 0$ if $w = X_i$ for some $i$ and $g(w) = \infty$ otherwise. This is equivalent to applying a prior $p(W_i)$ with finite support set on the rows of $X$ to each row of $W$. In addition, the regularization of $F$ can be transformed to geometric constraints between points in $X$ and in $X_A$. Since $F_{i,j}$ is the conical combination weight of $X_{A_j}$ in recovering $X_i$, a large $F_{i,j}$ intuitively indicates a small angle between $X_{A_j}$ and $X_i$, and vice verse. For example, the sparse and graph Laplacian prior for rows of $F$ in subspace clustering can be reduced to "cone clustering" for finding $A$. See [29] for an example of reducing the subspace clustering to general minimum conical hull problem.

## 3.2 Latent Variable Model

Different from deterministic MF, we build a system of equations from the moments of probabilistic latent variable models, and then formulate it as a general minimum conical hull problem, rather than directly solve it. Let the generalization model be $h \sim p(h; \alpha)$ and $x \sim p(x|h; \theta)$, where $h$ is a latent variable, $x$ stands for observation, and $\{\alpha, \theta\}$ are parameters. In a variety of graphical models such as GMMs and HMMs, we need to model conditional independence between groups of features. This is also known as the **multi-view assumption**. *W.l.o.g.*, we assume that $x$ is composed of three groups(views) of features $\{x_i\}_{i \in [3]}$ such that $\forall i \neq j, x_i \perp\!\!\!\perp x_j | h$. **We further assume the dimension $k$ of $h$ is smaller than** $p_i$, the dimension of $x_i$. Since the goal is learning $\{\alpha, \theta\}$, decomposing the moments of $x$ rather than the data matrix $X$ can help us get rid of the latent variable $h$ and thus avoid alternating minimization between $\{\alpha, \theta\}$ and $h$. When $\mathbb{E}(x_i|h) = h^T O_i^T$ (**linearity assumption**), the second and third order moments can be written in the form of matrix operator.

$$\begin{cases} \mathbb{E}(x_i \otimes x_j) = \mathbb{E}[\mathbb{E}(x_i|h) \otimes \mathbb{E}(x_j|h)] = O_i \mathbb{E}(h \otimes h) O_j^T, \\ \mathbb{E}(x_i \otimes x_j \cdot \langle \eta, x_l \rangle) = O_i \left[ \mathbb{E}(h \otimes h \otimes h) \times_3 (O_l \eta) \right] O_j^T, \end{cases} \quad (4)$$

where $A \times_n U$ denotes the $n$-mode product of a tensor $A$ by a matrix $U$, $\otimes$ is the outer product, and the operator parameter $\eta$ can be any vector. We will mainly focus on the models in which $\{\alpha, \theta\}$ can be exactly recovered from conditional mean vectors $\{O_i\}_{i \in [3]}$ and $\mathbb{E}(h \otimes h)$[1], because they cover most popular models such as GMMs and HMMs in real applications.

The left hand sides (LHS) of both equations in (4) can be directly estimated from training data, while their right hand sides (RHS) can be written in a unified matrix form $O_i D O_j^T$ with $O_i \in \mathbb{R}^{p_i \times k}$ and $D \in \mathbb{R}^{k \times k}$. By using different $\eta$, we can obtain $2 \leq q \leq p_l + 1$ independent equations, which compose a system of equations for $O_i$ and $O_j$. Given the LHS, we can obtain the column spaces of $O_i$ and $O_j$, which respectively equal to the column and row space of $O_i D O_j^T$, a low-rank matrix when $p_i > k$. In order to further determine $O_i$ and $O_j$, our discussion falls into two types of $D$.

**When $D$ is a diagonal matrix.** This happens when $\forall i \neq j, \mathbb{E}(h_i h_j) = 0$. A common example is that $h$ is a label/state indicator such that $h = e_i$ for class/state $i$, e.g., $h$ in GMM and HMM. In this case, the two $D$ matrices in the RHS of (4) are

$$\begin{cases} \mathbb{E}(h \otimes h) = \text{Diag}(\overrightarrow{\mathbb{E}(h_i^2)}), \\ \mathbb{E}(h \otimes h \otimes h) \times_3 (O_l \eta) = \text{Diag}(\overrightarrow{\mathbb{E}(h_i^3)} \cdot O_l \eta), \end{cases} \quad (5)$$

where $\overrightarrow{\mathbb{E}(h_i^t)} = [\mathbb{E}(h_1^t), \dots, \mathbb{E}(h_k^t)]$. So either matrix in the LHS of (4) can be written as a sum of $k$ rank-one matrices, i.e., $\sum_{s=1}^{k} \sigma^{(s)} O_i^s \otimes O_j^s$, where $O_i^s$ is the $s^{th}$ column of $O_i$.

The general separability assumption posits that the set of $k$ rank-one basis matrices constructing the RHS of (4) is a unique subset $A \subseteq [n]$ of the $n$ samples of $x_i \otimes x_j$ constructing the left hand sides, i.e., $O_i^s \otimes O_j^s = [x_i \otimes x_j]_{A_s} = X_{A_s,i} \otimes X_{A_s,j}$, the outer product of $x_i$ and $x_j$ in $(A_s)^{th}$ data point.

Therefore, by applying $q-1$ different $\eta$ to (4), we obtain the system of $q$ equations in the following form, where $Y^t$ is the estimate of the LHS of $t^{th}$ equation from training data.

$$\forall t \in [q], Y^{(t)} = \sum_{s=1}^{k} \sigma_{t,s} [x_i \otimes x_j]_{A_s} \Leftrightarrow [\mathrm{vec}(Y^{(t)})]_{t\in[q]} = \sigma[\mathrm{vec}(X_{t,i} \otimes X_{t,j})]_{t\in A}. \qquad (6)$$

The right equation in (6) is an equivalent matrix representation of the left one. Its LHS is a $q \times p_i p_j$ matrix, and its RHS is the product of a $q \times k$ matrix $\sigma$ and a $k \times p_i p_j$ matrix. By letting $X \leftarrow [\mathrm{vec}(Y^{(t)})]_{t\in[q]}$, $F \leftarrow \sigma$ and $Y \leftarrow [\mathrm{vec}(X_{t,i} \otimes X_{t,j})]_{t\in[n]}$, we can fit (6) to $X = FY_A$ in Definition 2. Therefore, learning $\{O_i\}_{i\in[3]}$ is reduced to selecting $k$ rank-one matrices from $\{X_{t,i} \otimes X_{t,j}\}_{t\in[n]}$ indexed by $A$ whose conical hull covers the $q$ matrices $\{Y^{(t)}\}_{t\in[q]}$. Given the anchor set $A$, we have $\hat{O}_i = X_{A,i}$ and $\hat{O}_j = X_{A,j}$ by assigning real data points indexed by $A$ to the columns of $O_i$ and $O_j$. Given $O_i$ and $O_j$, $\sigma$ can be estimated by solving (6). In many models, a few rows of $\sigma$ are sufficient to recover $\alpha$. See [29] for a practical acceleration trick based on matrix completion.

**When $D$ is a symmetric matrix with nonzero off-diagonal entries.** This happens in "admixture" models, e.g., $h$ can be a general binary vector $h \in \{0,1\}^k$ or a vector on the probability simplex, and the conditional mean $\mathbb{E}(x_i|h)$ is a mixture of columns in $O_i$. The most well known example is LDA, in which each document is generated by multiple topics.

We apply the general separability assumption by only using the first equation in (4), and treating the matrix in its LHS as $X$ in $X = FX_A$. When the data are extremely sparse, which is common in text data, selecting the rows of the denser second order moment as bases is a more reasonable and effective assumption compared to sparse data points. In this case, the $p$ rows of $F$ contain $k$ unit vectors $\{e_i\}_{i\in[k]}$. This leads to a natural assumption of "anchor word" for LDA [3].

See [29] for the example of reducing multi-view mixture model, HMM, and LDA to general minimum conical hull problem. It is also worth noting that we can show our method, when applied to LDA, yields equal results but is faster than a Bayesian inference method [3], see Theorem 4 in [29].

## 4 Algorithms for Minimum Conical Hull Problem

### 4.1 Divide-and-Conquer Anchoring (DCA) for General Minimum Conical Hull Problems

The key insights of DCA come from two observations on the geometry of the convex cone. First, projecting a conical hull to a lower-D hyperplane partially preserves its geometry. This enables us to distribute the original problem to a few much smaller sub-problems, each handled by a solver to the minimum conical hull problem. Secondly, there exists a very fast anchoring algorithm for a sub-problem on 2D plane, which only picks two anchor points based on their angles to an axis without iterative optimization or greedy pursuit. This results in a significantly efficient DCA algorithm that can be solely used, or embedded as a subroutine, checking if a point is covered in a conical hull.

### 4.2 Distributing Conical Hull Problem to Sub-problems in Low Dimensions

Due to the convexity of cones, a low-D projection of a conical hull is still a conical hull that covers the projections of the same points covered in the original conical hull, and generated by the projections of a subset of anchors on the extreme rays of the original conical hull.

**Lemma 2.** *For an arbitrary point $x \in cone(Y_A) \subset R^p$, where $A$ is the index set of the $k$ anchors (generators) selected from $Y$, for any $\Phi \in \mathbb{R}^{p \times d}$ with $d \leq p$, we have*

$$\exists \tilde{A} \subseteq A : x\Phi \in cone(Y_{\tilde{A}}\Phi), \qquad (7)$$

Since only a subset of $A$ remains as anchors after projection, solving a minimum conical hull problem on a single low-D hyperplane rarely returns all the anchors in $A$. However, the whole set $A$ can be recovered from the anchors detected on multiple low-D hyperplanes. By sampling the projection matrix $\Phi$ from a random ensemble $\mathbb{M}$, it can be proved that *w.h.p.* solving only $s = \mathcal{O}(ck \log k)$ sub-problems are sufficient to find all anchors in $A$. Note $c/k$ is the lower bound of angle $\alpha - 2\beta$ in Theorem 1, so large $c$ indicates a less flat conical hull. See [29] for our method's robustness to the failure in identifying "flat" anchors.

For the special case of NMF when $X = FX_A$, the above result is proven in [28]. However, the analysis cannot be trivially extended to the general conical hull problem when $X = FY_A$ (see Figure 1). A critical reason is that the converse of Lemma 2 does not hold: the uniqueness of the anchor set $\tilde{A}$

**Algorithm 1** DCA($X, Y, k, \mathbb{M}$)

---

**Input:** Two sets of points (rows) $X \in \mathbb{R}^{n \times p}$ and $Y \in \mathbb{R}^{m \times p}$ in matrix forms (ref. Table 1 to see $X$ and $Y$ for different models), number of latent factors/variables $k$, random matrix ensemble $\mathbb{M}$;
**Output:** Anchor set $A \subseteq [m]$ such that $\forall i \in [n], X_i \in cone(Y_A)$;
*Divide Step (in parallel):*
**for** $i = 1 \to s := \mathcal{O}(k \log k)$ **do**
    Randomly draw a matrix $\Phi \in \mathbb{R}^{p \times d}$ from $\mathbb{M}$;
    Solve sub-problem such as $\tilde{A}^t = \text{MCH}(X\Phi, Y\Phi)$ by any solver, e.g., (10);
**end for**
*Conquer Step:*
$\forall i \in [m]$, compute $\hat{g}(Y_i) = (1/s) \sum_{t=1}^{s} \mathbb{1}_{\tilde{A}^t}(Y_i)$;
Return $A$ as index set of the $k$ points with the largest $\hat{g}(Y_i)$.

---

on low-D hyperplane could be violated, because non-anchors in $Y$ may have non-zero probability to be projected as low-D anchors. Fortunately, we can achieve a unique $\tilde{A}$ by defining a "minimal conical hull" on a low-D hyperplane. Then Proposition 1 reveals when *w.h.p.* such an $\tilde{A}$ is a subset of $A$.

**Definition 4** (**Minimal conical hull**). *Given two sets of points (rows) $X$ and $Y$, the conical hull spanned by anchors (generators) $Y_A$ is the minimal conical hull covering all points in $X$ iff*

$$\forall \{i, j, s\} \in \left\{ i, j, s \mid i \in A^C = [m] \setminus A, j \in A, s \in [n], X_s \in cone(Y_A) \cap cone(Y_{i \cup (A \setminus j)}) \right\} \quad (8)$$

*we have $\widehat{X_s Y_i} > \widehat{X_s Y_j}$, where $\widehat{xy}$ denotes the angle between two vectors $x$ and $y$. The solution of minimal conical hull is denoted by $A = \text{MCH}(X, Y)$.*

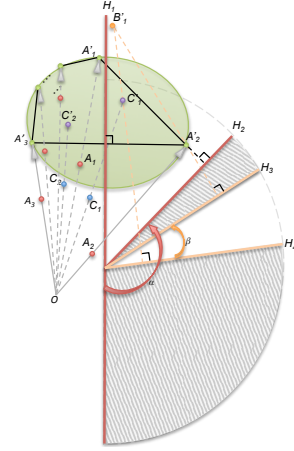

It is easy to verify that the minimal conical hull is unique, and the general minimum conical hull problem $X = FY_A$ under the general separability assumption (which leads to the identifiability of $A$) is a special case of $A = \text{MCH}(X, Y)$. In DCA, on each low-D hyperplane $H_i$, the associated sub-problem aims to find the anchor set $\tilde{A}^i = \text{MCH}(X\Phi^i, Y\Phi^i)$. The following proposition gives the probability of $\tilde{A}^i \subseteq A$ in a sub-problem solution.

**Proposition 1** (**Probability of success in sub-problem**). *As defined in Figure 2, $A_i \in A$ signifies an anchor point in $Y_A$, $C_i \in X$ signifies a point in $X \in \mathbb{R}^{n \times p}$, $B_i \in A^C$ signifies a non-anchor point in $Y \in \mathbb{R}^{m \times p}$, the green ellipse marks the intersection hyperplane between $cone(Y_A)$ and the unit sphere $\mathbb{S}^{p-1}$, the superscript $\cdot'$ denotes the projection of a point on the intersection hyperplane. Define d-dim ($d \leq p$) hyperplanes $\{H_i\}_{i \in [4]}$ such that*

Figure 2: Proposition 1.

$A_3' A_2' \perp H_1$, $A_1' A_2' \perp H_2$, $B_1' A_2' \perp H_3$, $B_1' C_1' \perp H_4$, *let $\alpha = \widehat{H_1 H_2}$ be the angle between hyperplanes $H_1$ and $H_2$, $\beta = \widehat{H_3 H_4}$ be the angle between $H_3$ and $H_4$. If $H$ with associated projection matrix $\Phi \in \mathbb{R}^{p \times d}$ is a d-dim hyperplane uniformly drawn from the Grassmannian manifold $\text{Gr}(d, p)$, and $\tilde{A} = MCH(X\Phi, Y\Phi)$ is the solution to the minimal conical hull problem, we have*

$$\Pr(B_1 \in \tilde{A}) = \frac{\beta}{2\pi}, \Pr(A_2 \in \tilde{A}) = \frac{\alpha - \beta}{2\pi}. \quad (9)$$

See [29] for proof, discussion and analysis of robustness to unimportant "flat" anchors and data noise.

**Theorem 1** (**Probability bound**). *Following the same notations in Proposition 1, suppose $p^{**} = \min_{\{A_1, A_2, A_3, B_1, C_1\}} (\alpha - 2\beta) \geq c/k > 0$. It holds with probability at least $1 - k \exp\left(-\frac{cs}{3k}\right)$ that DCA successfully identifies all the $k$ anchors in $A$, where $s$ is the number of sub-problems solved.*

See [29] for proof. Given Theorem 1, we can immediately achieve the following corollary about the number of sub-problems that guarantee success of DCA in finding $A$.

**Corollary 1** (**Number of sub-problems**). *With probability $1 - \delta$, DCA can correctly recover the anchor set $A$ by solving $\Omega(\frac{3k}{c} \log \frac{k}{\delta})$ sub-problems.*

See [29] for the idea of divide-and-conquer randomization in DCA, and its advantage over *Johnson-Lindenstrauss (JL) Lemma* based methods.

## 4.3 Anchoring on 2D Plane

Although DCA can invoke any solver for the sub-problem on any low-D hyperplane, a very fast solver for the 2D sub-problem always shows high accuracy in locating anchors when embedded into DCA. Its motivation comes from the geometry of conical hull on a 2D plane, which is a special case of a $d$-dim hyperplane $H$ in the sub-problem of DCA. It leads to a non-iterative algorithm for $A = \mathrm{MCH}(X, Y)$ on the 2D plane. It only requires computing $n + m$ cosine values, finding the min/max of the $n$ values, and comparing the remaining $m$ ones with the min/max value.

According to Figure 1, the two anchors $Y_{\tilde{A}}\Phi$ on a 2D plane have the min/max (among points in $Y\Phi$) angle (to either axis) that is larger/smaller than all angles of points in $X\Phi$, respectively. This leads to the following closed form of $\tilde{A}$.

$$\tilde{A} = \{\arg\min_{i \in [m]} ((\widehat{Y_i\Phi})\varphi - \max_{j \in [n]} (\widehat{X_j\Phi})\varphi)_+, \arg\min_{i \in [m]} (\min_{j \in [n]} (\widehat{X_j\Phi})\varphi - (\widehat{Y_i\Phi})\varphi)_+\}, \qquad (10)$$

where $(x)_+ = x$ if $x \geq 0$ and $\infty$ otherwise, and $\varphi$ can be either the vertical or horizontal axis on a 2D plane. By plugging (10) in DCA as the solver for $s$ sub-problems on random 2D planes, we can obtain an extremely fast learning algorithm.

Note for the special case when $X = Y$, (10) degenerates to finding the two points in $X\Phi$ with the smallest and largest angles to an axis $\varphi$, i.e., $\tilde{A} = \{\arg\min_{i \in [n]} (\widehat{X_i\Phi})\varphi, \arg\max_{i \in [n]} (\widehat{X_i\Phi})\varphi\}$. This is used in matrix factorization and the latent variable model with nonzero off-diagonal $D$.

See [29] for embedding DCA as a fast subroutine into other methods, and detailed off-the-shelf DCA algorithms of NMF, SC, GMM, HMM and LDA. A brief summary is in Table 1.

## 5 Experiments

See [29] for a complete experimental section with results of DCA for NMF, SC, GMM, HMM, and LDA, and comparison to other methods on more synthetic and real datasets.

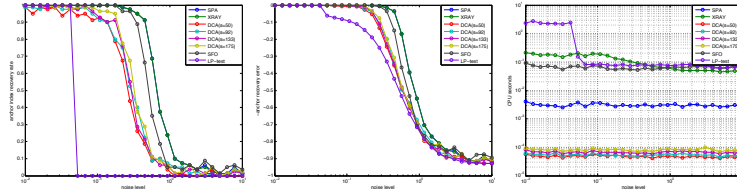

Figure 3: Separable NMF on a randomly generated $300 \times 500$ matrix, each point on each curve is the result by averaging 10 independent random trials. SFO-greedy algorithm for submodular set cover problem. LP-test is the backward removal algorithm from [4]. LEFT: Accuracy of anchor detection (higher is better). Middle: Negative relative $\ell_2$ recovery error of anchors (higher is better). Right: CPU seconds.

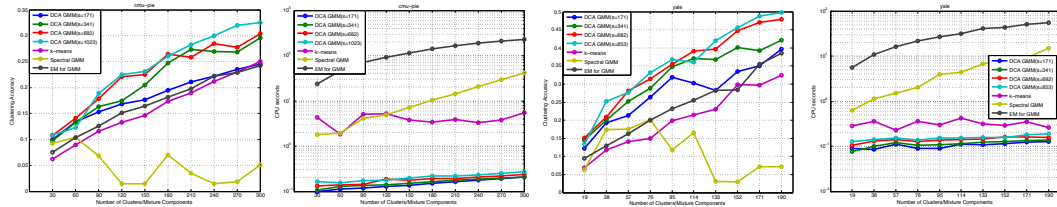

Figure 4: Clustering accuracy (higher is better) and CPU seconds vs. Number of clusters for Gaussian mixture model on CMU-PIE (left) and YALE (right) human face dataset. We randomly split the raw pixel features into 3 groups, each associates to a view in our multi-view model.

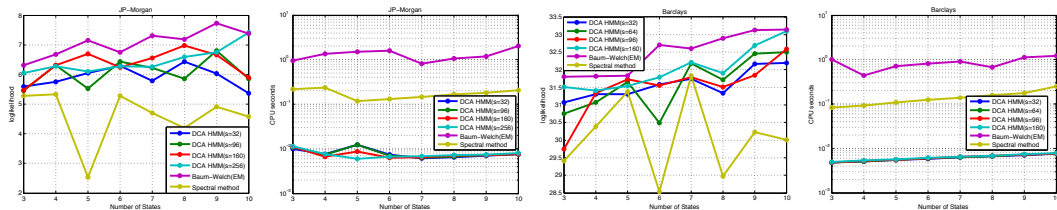

Figure 5: Likelihood (higher is better) and CPU seconds vs. Number of states for using an HMM to model the stock price of 2 companies from 01/01/1995-05/18/2014 collected by Yahoo Finance. Since no ground truth label is given, we measure likelihood on training data.

**DCA for Non-negative Matrix Factorization on Synthetic Data**. The experimental comparison results are shown in Figure 3. Greedy algorithms SPA [14], XRAY [19] and SFO achieves the best

accuracy and smallest recovery error when noise level is above 0.2, but XRAY and SFO are the slowest two. SPA is slightly faster but still much slower than DCA. DCA with different number of sub-problems shows slightly less accuracy than greedy algorithms, but the difference is acceptable. Considering its significant acceleration, DCA offers an advantageous trade-off. LP-test [4] has the exact solution guarantee, but it is not robust to noise, and too slow. Therefore, DCA provides a much faster and more practical NMF algorithm with comparable performance to the best ones.

**DCA for Gaussian Mixture Model on CMU-PIE and YALE Face Dataset**. The experimental comparison results are shown in Figure 4. DCA consistently outperforms other methods (k-means, EM, spectral method [1]) on accuracy, and shows speedups in the range 20-2000. By increasing the number of sub-problems, the accuracy of DCA improves. Note the pixels of face images always exceed 1000, and thus results in slow computation of pairwise distances required by other clustering methods. DCA exhibits the fastest speed because the number of sub-problems $s = \mathcal{O}(k \log k)$ does not depend on the feature dimension, and thus merely 171 2D random projections are sufficient for obtaining a promising clustering result. The spectral method performs poorer than DCA due to the large variance of sample moments. DCA uses the separability assumption in estimating the eigenspace of the moment, and thus effectively reduces the variance.

Table 2: Motion prediction accuracy (higher is better) of the test set for 6 motion capture sequences from CMU-mocap dataset. The motion for each frame is manually labeled by the authors of [16]. In the table, s13s29(39/63) means that we split sequence 29 of subject 13 into sub-sequences, each has 63 frames, in which the first 39 ones are used for training and the rest are for test. Time is measured in ms.

| Sequence | s13s29(39/63) | | s13s30(25/51) | | s13s31(25/50) | | s14s06(24/40) | | s14s14(29/43) | | s14s20(29/43) | |
|---|---|---|---|---|---|---|---|---|---|---|---|---|
| Measure | Acc | Time | Acc | Time | Acc | Time | Acc | Time | Acc | Time | Acc | Time |
| Baum-Welch (EM) | 0.50 | 383 | 0.50 | 140 | 0.46 | 148 | 0.34 | 368 | 0.62 | 529 | 0.77 | 345 |
| Spectral Method | 0.20 | 80 | 0.25 | 43 | 0.13 | 58 | 0.29 | 66 | 0.63 | 134 | 0.59 | 70 |
| DCA-HMM (s=9) | 0.33 | 3.3 | 0.92 | 1 | 0.19 | 1.5 | 0.29 | 4.8 | 0.79 | 3 | 0.28 | 3 |
| DCA-HMM (s=26) | 0.50 | 3.3 | **1.00** | 1 | **0.65** | 1.6 | **0.60** | 4.8 | 0.45 | 3 | **0.89** | 3 |
| DCA-HMM (s=52) | 0.50 | 3.4 | 0.50 | 1.1 | 0.43 | 1.6 | 0.48 | 4.9 | **0.80** | 3.2 | 0.78 | 3.1 |
| DCA-HMM (s=78) | **0.66** | 3.4 | 0.93 | 1.1 | 0.41 | 1.6 | 0.51 | 4.9 | **0.80** | 6.7 | 0.83 | 3.2 |

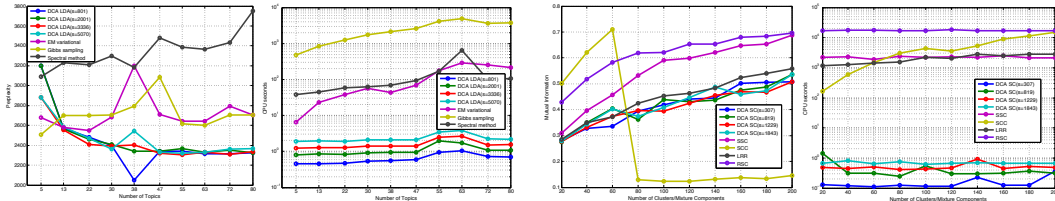

Figure 6: **LEFT:** Perplexity (smaller is better) on test set and CPU seconds vs. Number of topics for LDA on NIPS1-17 Dataset, we randomly selected 70% documents for training and the rest 30% is used for test. **RIGHT:** Mutual Information (higher is better) and CPU seconds vs. Number of clusters for subspace clustering on COIL-100 Dataset.

**DCA for Hidden Markov Model on Stock Price and Motion Capture Data**. The experimental comparison results for stock price modeling and motion segmentation are shown in Figure 5 and Table 2, respectively. In the former one, DCA always achieves slightly lower but comparable likelihood compared to Baum-Welch (EM) method [5], while the spectral method [2] performs worse and unstably. DCA shows a significant speed advantage compared to others, and thus is more preferable in practice. In the latter one, we evaluate the prediction accuracy on the test set, so the regularization caused by separability assumption leads to the highest accuracy and fastest speed of DCA.

**DCA for Latent Dirichlet Allocation on NIPS1-17 Dataset**. The experimental comparison results for topic modeling are shown in Figure 6. Compared to both traditional EM and the Gibbs sampling [23], DCA not only achieves both the smallest perplexity (highest likelihood) on the test set and the highest speed, but also the most stable performance when increasing the number of topics. In addition, the "anchor word" achieved by DCA provides more interpretable topics than other methods.

**DCA for Subspace Clustering on COIL-100 Dataset**. The experimental comparison results for subspace clustering are shown in Figure 6. DCA provides a much more practical algorithm that can achieve comparable mutual information but at a more than 1000 times speedup over the state-of-the-art SC algorithms such as SCC [9], SSC [12], LRR [21], and RSC [26].

**Acknowledgments**: We would like to thank MELODI lab members for proof-reading and the anonymous reviewers for their helpful comments. This work is supported by TerraSwarm research center administered by the STARnet phase of the Focus Center Research Program (FCRP) sponsored by MARCO and DARPA, by the National Science Foundation under Grant No. (IIS-1162606), and by Google, Microsoft, and Intel research awards, and by the Intel Science and Technology Center for Pervasive Computing.

## Footnotes

[1]Note our method can also handle more complex models that violate the linearity assumption and need higher order moments for parameter estimation. By replacing $x_i$ in (4) with $\text{vec}(x_i \otimes^n)$, the vectorization of the $n^{th}$ tensor power of $x_i$, $O_i$ can contain $n^{th}$ order moments for $p(x_i|h; \theta)$. However, since higher order moments are either not necessary or difficult to estimate due to high sample complexity, we will not study them in this paper.

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
