[Reviews · NeurIPS 2014]

Submitted by Assigned_Reviewer_1

This paper reduces a broad class of machine learning problems involving latent variables to the problem of finding "anchors" defining the conical hull of the data (via the method of moments). In addition, it proposes a new divide-and-conquer algorithm based on random projections to speed up the search for the anchors.

Overall, I found this an interesting paper presenting significant contributions. However the presentation could be greatly improved as it lacks clarity here and there. It looks like this paper was squeezed in a hurry to fit the 8-page limit.

In Definition 2, line 113, isn't the identity matrix only for the case X=Y?

Line 122 is not very precise: "finds the subset of rows in Y that define the same cone as all the rows in X" -- it's not necessarily the same cone, but a super-cone, i.e., one which contains cone(X).

Line 132: "in the center" -- which center?

Line 244: why is [4] a "Bayesian learning method"?

Definition 4 is confusing. What are i,j,s representing?

Figures 3, 4, 5, 6 are too small and the font size in the legend is tiny. As it stands it's unreadable.

A conclusion section is missing.
Summary: This paper reduces a broad class of machine learning problems involving latent variables to the problem of finding "anchors" defining the conical hull of the data (via the method of moments). In addition, it proposes a new divide-and-conquer algorithm based on random projections to speed up the search for the anchors.
Overall, I found this an interesting paper presenting significant contributions. However the presentation could be greatly improved as it lacks clarity here and there. It looks like this paper was squeezed in a hurry to fit the 8-page limit.

Submitted by Assigned_Reviewer_18

Summary: This work proposes a new algorithm, "divide-and-conquer
anchoring" (DCA), that solves the problem of selecting a subset A of
rows from a matrix Y such that those rows best model another matrix X:
X = FY_A. Previous work [29] addresses the simpler setting where X =
Y and shows how this setting generalizes non-negative matrix
factorization (NMF). The even more general X != Y setting that this
work tackles is shown to generalize many core machine learning
problems: Gaussian mixture models (GMMs), hidden Markov models (HMMs),
latent Dirichlet allocation (LDA), and subspace clustering (SC). This
setting relies on a "general separability assumption", which requires
that there exist a subset A of Y's rows such that the rows of X are
contained in their cone. Lemma 1 gives a very simple condition for
the identifiability of A, which allows DCA to be much more robust to
variance in the sample moments than spectral methods. Experimental
results on a large number of tasks indicate that DCA is almost always
much faster than traditional learning methods (including spectral
ones) without losing significant accuracy.

Comments:

Clarity, quality: The general idea and algorithm of the paper is
clear, but the math in this work is pretty dense. Additional small,
illustrative examples and figures (similar to that of Figure 2) would
be helpful to make some of the concepts more concrete.

Originality: The work of [29] seems to be the most closely related to
this paper. The X != Y difference represents a large delta in terms
of theoretical and experimental work though, so this paper is
certainly sufficiently original.

Significance: Given the large number of practical machine learning
tasks that this minimum conical hull problem encompasses, and its
clear performance advantages on them, this work seems very
significant.

Supplement: It's somewhat difficult to map back and forth between the
document and the supplement, since the theorem/lemma/etc. numbers
don't match. When referring to [3] (the supplement) from the main
paper, it would be helpful if the authors could use the optional
argument of the cite command to indicate more precisely which part of
[3] is being referenced: \cite[specific location]{supplement}.
Additionally, there is a lot of information in the supplement that
isn't represented in the main body of the paper. Perhaps this paper
would be better suited to a venue without page limits (such as a
journal).

Experiments: Do the authors have plans to release the associated code?
How were the s-values (50, 92, 133, 175) selected? Often DCA, wins
speed-wise but loses slightly accuracy-wise. Can the authors comment
further on why this might be the case? Are there knobs that can but
turned on the DCA algorithm to alter the speed/accuracy balance?
(Besides the s-knob, which doesn't seem to have a huge effect.)

Organization: Since NMF is already covered by [29], it might make
sense to leave its explanation and experiments out of the main body of
the paper and focus on explaining the other tasks (or the theory) in
greater detail.

The variable "c": The variable "c" in the definition of "s" on line
268 is only really defined in the supplement. Even there it's
somewhat unclear what c's range is. Can the authors provide a
mathematical statement describing the relationship between c and the
flatness of the flattest anchor? Is n the maximum value for c?

Projection matrices: Proposition 1 indicates that projection matrices
are drawn uniformly from the Grassmanian manifold, and the supplement
indicates several other random matrix ensembles that could be drawn
from. It would be helpful if the authors could provide a few
references detailing how to draw from each of these.

More minor notes:

1) "although spectral method using" -> "although spectral methods
using"

2) "random drawn low-dimensional" -> "randomly drawn low-dimensional"

3) Equation (1) and elsewhere: Use \left and \right to make brackets
and parentheses match the height of the math that they enclose.

2) "general models such as matrix factorization and latent variable
model" -> "general models such as matrix factorization and latent
variable models"

4) "most popular models such as GMM and HMM" -> "most popular models
such as GMMs and HMMs"

5) "the two D matrices in RHS of" -> "the two D matrices in the RHS of"

6) "either matrix in LHS of" -> "either matrix in the LHS of"

7) "each handled by a solver to minimum conical hull problem" -> "each
handled by a solver to the minimum conical hull problem"

8) "anchoring algorithm for sub-problem on 2D-plane" -> "anchoring
algorithm for a sub-problem on the 2D-plane"

9) "Due to the convexity of cone" -> "Due to the convexity of cones"

10) "For arbitrary point" -> "For an arbitrary point"

11) "w.h.p. solving" -> "w.h.p.\ solving"

12) "given two sets of points(rows)" -> "given two sets of points
(rows)"

13) "under general separability assumption" -> "under the general
separability assumption"

14) Supplement, line 415: "Lemma ??" -> "Lemma 2"

15) "each point on each curve is the result by averaging" -> "each
point on each curve is the result of averaging"

16) "Because DCA uses separability assumption" -> "DCA uses the
separability assumption"

17) There are many grammatical errors in the supplement besides the
ones listed above. It could use a more careful proofreading.
Summary: This work proposes a new algorithm, "divide-and-conquer anchoring"
(DCA), that solves a problem which generalizes many other core machine
learning problems (NMF, GMMS, HMMs, LDA, SC). It is both interesting
theoretically and performs well experimentally.

Submitted by Assigned_Reviewer_33

This paper proposes to solve the problem of finding k extremal rays of minimum conical hull, as an alternative to common algorithms such as EM. In order to reach the solutions, the authors introduce a divide-and-conquer scheme to reduce the original problem into sub-problems that are easy to be solved. Applied on several models, this method shows competitive performances.

The paper is overall clear. The good experimental results and simplicity of implementation suggest the usefulness of the proposed method. The supplementary contains enough details.

The originality is limited, since it combines several tricks that have been widely used in the optimization and machine learning societies, such as the conical hull formulation and the random projection trick.

Although the work is technically correct, the following points would require clarification:

- The notations should be checked for the reading proficiency. There are notations without specifications. For example, in Line 138, there is no specification for the operator "otimes", which is inappropriately located at Line 191.

- More literature reviews on using conical hull for parameter optimizations are expected to improve the clarity of this work.

- Concerning the experimental results, more methods are expected to be included in the comparison. For instance, the Viterbi (MAP) training for HMM would better to be included besides the Baum-Welch (EM) algorithm. Moreover, the computational cost of k-means GMM learning seems weird. In contrast to EM, why the time cost of k-means almost holds the same as the number of components increases? If there is some trick for k-means, it needs specifications.
Summary: The work overall is interesting and appears to work well empirically. As it stands, I think the theoretical analysis is incomplete and some missed content is needed.
Author Feedback
Author rebuttal: We would like to thank all the reviewers for their efforts in reviewing this paper and constructive comments!

Re Reviewer #1:

Yes it is not easy to squeeze the 23 page full version to only 8 pages, but we will try our best to make the short version clearer than current one, remove all the typos, adjust the font sizes in figures, and add a conclusion session (one is available in our Arxiv version). For the current draft, all the missing details are contained in the submitted supplemental material.

Thanks for the comments about line 113 and 122! Yes they are only under the separability assumption and should be changed in the general version. But it is worth noting that these two typos will not change the correctness of the other parts of this paper.

Line 132: in the center of multiple layers of conical hulls.

Line 244: it should be a “Bayes’ learning method”, because it applies Bayes rules in formulating the matrix factorization model of probabilities, and recover the topic probability from matrix factorization by using Bayes’ rule.

Definition 4: i, j, s are defined in the conditions in (8), s indexes arbitrary point in X, j indexes arbitrary point in subset A of Y, i indexes arbitrary point in the complementary set of subset A of Y.

Re Reviewer #18:

Thanks for your positive comments about our work, and the detailed minor notes! We find they are very useful, and will use them for the revision. We will also improve the mapping between the short version and long version, as well as the organization. The code for all different models will be released later.

The number of sub-problems s can be selected according to the theoretical bound, i.e., s=O(klogk), and computational budget. The {50, 92, 133, 175} in NMF experiment is just a randomly generated increasing sequence of s showing how the performance and speed will change when increasing s. On NMF, DCA loses a slight accuracy because it uses a randomized strategy (others are deterministic), and we evaluate recovery accuracy on training set. When s goes to infinity, it is expected to achieve the same performance as others. It worth noting that we rarely see this slight drawback on other tasks, which instead evaluates generative performance on a test set. Except s, you can also adjust the dimension of each sub-problem to gain a speed/accuracy trade-off. But in this case, the sub-problems are hard to be solved by our fast anchoring algorithm, and slower iterative algorithms are needed.

Variable “c”: by the beginning of Theorem 3, c/k (k is the number of anchors) is defined as the lower bound for \alpha-2\beta, where \alpha and \beta are angles defined in (27). From Figure 2, large c comes with large \alpah and small \beta, which indicate a less “flat” anchor.

The projection matrix can be some fast random transformation or random sparse matrix. We will include references to them in later version.

Re Reviewer #33:

Thanks for your comments! However, we do not think our work is a simple combination of existing tricks. We declared in our paper (line 057-061) our major difference, compared to the conical hull problem, is that we use model X=FY_A rather than X=FX_A. This requires us to build new theory for the new model (Definition 2 & 3, Lemma 1), that allows us to apply our model to a much richer class of learning problems that cannot be addressed by existing conical hull methods. In addition, different from JL lemma based random projection (RP), we use RP to only preserve partial information and thus it could be of ultra low (1 or 2) dimensions. This is benefited from the new “divide-and-conquer randomization” idea (line 076-078), which is different from most current “batch” RP methods. New theory has been built for this idea as well (Proposition 1, Theorem 1, Corollary 1).

We will improve the notation and make them clearer in the final version, however, given the chance.

We will also include more baseline methods in our empirical comparisons. The computational cost of k-means in our paper is less than normal k-means because we use a faster public available implementation called “litekmeans”.

We reviewed six papers ([4, 5, 12, 15, 20, 29]) about conical/convex hull problems for parameter optimization, all state-of-the-art. We also provided empirical comparison to most of them. Given the one-page limit of references, and because one of our major contributions is how to reduce other problems to conical hull problem rather than only developing fast conical hull algorithm, we believe these are sufficient. However, we will include more in the long version of the paper.

The optimization problem as well as object of general conical hull problem has been formally given in (3). The objective equals to solve equation X=FY_A, or to minimize \|X-FY_A\|, which have been declared in Section 2. For matrix factorization, it is recovery error minimization. For latent variable models, the object is to fit moments by model parameters. This has been given in (4).